# Hybrid Therapeutic Device (CUHK-OA-M2) for Relieving Symptoms Induced by Knee Osteoarthritis

**DOI:** 10.3390/bioengineering10010095

**Published:** 2023-01-11

**Authors:** Li Zou, Kisum Chu, Xuan He, Ye Li, Liangbin Zhou, Xiayi Xu, Wei-Hsin Liao, Ling Qin

**Affiliations:** 1Department of Orthopaedics and Traumatology, The Chinese University of Hong Kong, Hong Kong, China; 2Department of Mechanical and Automation Engineering, The Chinese University of Hong Kong, Hong Kong, China; 3Hong Kong Centre for Cerebro-Cardiovascular Health Engineering, Hong Kong, China

**Keywords:** knee osteoarthritis, physical therapy, low-level laser, heat therapy, massage, physical stimulation

## Abstract

The symptoms of knee osteoarthritis (KOA) severely affect the life quality of the elderly population. Low-level laser therapy, heat therapy, and massage therapy are widely used as independent treatments for joint disorders. However, there are very limited reports of a combination of these therapies into an integrated device for KOA so far. This study aims to develop a novel hybrid therapeutic device that can meet various requirements for knee therapy. Our hybrid therapeutic device (CUHK-OA-M2) integrated with low-level laser therapy, heat therapy, and local massage therapy can effectively provide patients with KOA with relief from their clinical symptoms. A pilot test of 50 community-dwelling elderly volunteers with KOA was performed. Finally, 43 volunteers completed two treatment periods (30 days each) and two post-treatment periods (30 days each). The Western Ontario and McMaster Universities Osteoarthritis Index (WOMAC) scores were collected and analyzed after each period. The outputs of the low-level laser, heating, and massage therapies significantly decreased the WOMAC scores in terms of pain, stiffness, function and total WOMAC after two treatment periods (*p* < 0.05). Although the score increased slightly after the post-treatment period, it was still lower than the baseline, indicating the treatment outcome could last for an extended period. Therefore, our CUHK-OA-M2 device, as an integrated multi-functional hybrid therapeutic device, is therapeutically significant for treating osteoarthritis symptoms on the knee joints of elderly subjects.

## 1. Introduction

Osteoarthritis (OA) can be a consequence of chronic musculoskeletal diseases, trauma, or aging-related degeneration [1,2,3]. The knee, as the major weight-bearing joint, has high morbidity of OA. Clinically, symptoms such as joint pain, stiffness, limited joint motion, and joint effusion are usually experienced by patients at all stages of knee OA (KOA) [4,5]. As patients suffer these unpleasant symptoms, their life quality and social activities are significantly affected, which further leads to chronic and systematic diseases or mental depression [6]. Total knee arthroplasty is regarded as the final surgical solution for patients experiencing intolerable symptoms. For most patients, especially those at an early stage of KOA, non-surgical treatments are feasible and recommended [7,8,9]. According to the clinical guidelines, mind–body management, physical therapies, and pharmacological interventions are indicated to relieve KOA symptoms [10]. In some situations, acupuncture has also been advised to relieve joint pain [11]. In recent years, low-level laser therapy has been increasingly used to enhance the treatment effects of OA. Many studies have reported that low-level laser therapy can stimulate the oxidative respiratory chain to boost tissue metabolism [12,13,14,15,16,17,18]. Heat therapy has been highly recommended to relieve OA symptoms within a short period [19,20]. Massage therapy has also been reported to relieve the muscle tension around the joint to attenuate physical and mental suffering [21,22,23].

Laser therapy devices, heat therapy devices, and massage devices have been developed separately and used widely in communities and hospitals [24,25]. While these treatment devices can alleviate uncomfortable feelings and improve knee function to a certain degree [26], they still have many limitations. Firstly, most devices are designed with a single biophysical parameter for treatment. Secondly, the biophysical instruments used in hospitals are usually expensive and require a large space for their installation and application [19,20], which indicates that individual patients are not able to afford them in home-based facilities. Therefore, portable devices that contain multiple therapies with more parameters are highly desirable for daily rehabilitation [27,28,29,30]. Thirdly, most of the registered devices for knee joint applications are focused on the anterior side of the knee (kneecap) for easy fixation and wear. According to the knee joint anatomy shown in Figure 1, the main arteries, veins, lymph system, and nerves pass through the knee joint’s posterior side, known as the popliteal fossa [31,32,33]. Therefore, it is quite difficult for the signals from current anterior-side devices used in interventions, such as laser, heat, or massage therapies, to penetrate the patella and cruciate ligaments to reach the popliteal fossa. Technically, the sophisticated anatomy and kinetics of the posterior knee hinder the development of devices because it is extremely difficult to target the popliteal fossa directly.

To overcome above-mentioned limitations, we developed a medical device (named “CUHK-OA-M2”) for the treatment of the popliteal fossa. The device can perform low-level laser therapy (LLLT), heat therapy, and massage therapy simultaneously to achieve a combined therapy outcome to maximize treatment efficacy (Figure 2). In this study, we first tested the technical parameters and then assessed the treatment effects by conducting a pilot clinical test. We concluded that our hybrid therapeutic device can generate effective biophysical signals to significantly relieve the symptoms of KOA.

## 2. Materials and Methodology

### 2.1. Design of the Hybrid Therapeutic Device

As shown in Figure 2, we designed an entire biophysical system (CUHK-OA-M2) for KOA in the popliteal fossa in patients with early-stage KOA [4,6]. The therapy core unit was fixed inside the popliteal fossa by an adapter, where a heat therapy system (red area) and electrical system (blue area and green area) were integrated into the adapter. The massage therapy as mechanical stimulation and LLLT were integrated into the therapeutic device’s core. Driven by an actuator inside the core component, the rotation of outer rollers produced massage therapy. LLLT was implemented via a set of laser modules inside the core component.

The mechanical design of the core component is shown in Figure 2B. The device was composed of three crucial modules. The first module consisted of three main components: an actuator with five rollers (type A and type B) for massage therapy; two types of laser models with distinct wavelengths: 660 nm (T2835, output power: 0.4 W) and 850 nm (T2835, output power: 1.2 W); and a heating wire (nickel–chromium alloy) and a cotton cloth. These modules are summarized in Table 1 and the major parameters are listed in Table 2.

### 2.2. Control System of the Hybrid Therapeutic Device

The principle of the whole therapeutic device system and the application procedure are summarized in Figure 3A. Firstly, the user receives our combination treatment by wearing the proposed therapeutic device. The user can adjust the adapter of the therapeutic device to reach a satisfactory degree of tightness and start the treatment of the device. Then, based on the targeted speed, torque, and pressure force of the massage therapy, the controller sends the appropriate voltage signal to the actuator. The temperature is maintained within 40–42 °C by controlling the output current magnitude. Finally, the system provides sufficient optical power to the popliteal fossa to meet the therapeutic power density and dose value.

As shown in Figure 3B, in order to consume less electrical power in the warming system, we use a high current value at the beginning until the temperature of the skin surface reaches 41 °C and then use the second gear of the current level for the rest of the treatment. To avoid the temperature exceeding 43 °C, which is outside of the range of the combination treatment, warming system operations can be turned off during the treatment as shown in Figure 3B. With this on–off control methodology, the temperature of the skin surface can be maintained around 41 °C with suitable fluctuations.

### 2.3. Lab Experiment Setup for Massage Therapy, Heat Therapy, and LLLT

As shown in Figure 4, the prototype of the hybrid therapeutic device is worn on the knee joint, a torque-sensing system is applied to observe the work cycle of the actuator, a measurement system is designed to measure the result of massage force and heat therapy, and the power system supplies power to all of these electronics. The measurement system was composed of three parts, including a thermometer, a force-sensitive resistor, and the MCU. An optical power sensor was applied to measure the laser power of the LLLT module. The average laser energy was obtained from the calibration and calculation system.

### 2.4. Experimental Setup and Development

The experimental setup of the hybrid therapeutic device was established in the research laboratory as shown in Figure 5. The acupressure force could be changed by adjusting the bandage around the knee joint, and the force sensor and torque sensor determined the relationship between the preload and massage force. As shown in Figure 5B, the prototype of the measurement system was developed. The thermometer (B57164) is a kind of thermistor that responds to the temperature in real time. The force-sensitive resistor (FSR 402) is a unit to measure the pressure force and translate it into an electrical signal, and the MCU (Arduino Uno) is used to analyze the signals from the sensors. As shown by the LLLT test setup in Figure 5C, a sensor (VLP-2W, Beijing Ranbond Technology Co., Ltd., Beijing, China) was selected for laser power experiment.

### 2.5. Pilot Test on Elderly Subjects with OA Symptoms

#### 2.5.1. Inclusion and Exclusion

Community-dwelling elderly people aged 65 to 90 with KOA symptoms according to the standard of OARSI [4] were recruited from local community centers in Hong Kong. The exclusion criteria included patients at KOA stage 4 with lower limb malalignment, total knee arthroplasties, chronic spine diseases, peri/central nerve diseases, and dermatological diseases/allergies.

#### 2.5.2. Enrollment, Eligibility Assessment, and Grouping

After being invited to complete a form on informed consent, a total of 80 volunteers participated the pilot test in Hong Kong, but after eligibility assessment, 17 volunteers were excluded (15 volunteers were diagnosed with stage 4 KOA; 2 volunteers underwent TKA). The 63 eligible volunteers were allocated into one group. At the beginning of the trial, 13 volunteers quit when they were asked if they were willing to continue. Finally, 50 volunteers were included in the trial. Data from 43 volunteers were collected (contact was lost with 7 volunteers).

#### 2.5.3. Intervention Protocol

After allocation, study subjects consented to be treated with the CUHK-OA-M2 device for two treatment periods (30 days each). The combination function of the device included medically approved safe margin, including low-level laser therapy, heat therapy, and popliteal fossa massage therapy, which would be applied simultaneously. According to the test results of the device, the duration of treatment was 20 min, and the frequency was one treatment every 3 days. Then, all the volunteers entered two post treatment periods (30 days each). At days 30, 60, 90, and 120, all the WOMAC scores [34] (total score, pain score, stiffness score, and function score) were collected.

#### 2.5.4. Evaluation of Pain and Knee Function

The WOMAC consisted of 24 items divided into three subgroups: (1) Pain while walking, using stairs, in bed, sitting or lying down, and standing upright; (2) Stiffness after first waking and later in the day; (3) Physical Functioning while using stairs, getting up from a sitting position, standing, bending over, walking, getting in/out of a car, shopping, putting on/taking off socks, getting out of bed, lying in bed, getting in/out of the bath, sitting, getting on/off the toilet, heavy domestic duties, and light domestic duties. The questionnaire forms were collected on paper, over the phone, or by email. The questions were scored on a scale of 0–10, from none to extreme. The scores for each subscale were separately calculated. Finally, summing up the scores for all the subscales yielded a total WOMAC score. Higher scores indicated worse pain, stiffness, and functional limitations (http://www.rheumatology.org/practice/clinical/clinicianresearchers/outcomes-instrumentation/WOMAC.asp, accessed on 1 September 2021).

### 2.6. Statistical Analysis

Data were analyzed using SPSS 16.0 (SPSS Inc., Chicago, IL, USA) and GraphPad software 9.0 (GraphPad Software Inc., San Diego, CA, USA). One-way ANOVA and Tukey’s test (post hoc) were used for the primary outcome measure (WOMAC index). Resulting baseline-adjusted treatment effects are given together with 95% CI and corresponding *p*-values as well as means and standard deviation (SD) of the primary outcome for each time point. A *p*-value less than 0.05 was identified as having statistical significance.

## 3. Results

### 3.1. The CUHK-OA-M2 Delivered Effective Massage Force, Heating, and Laser Energy

As shown in Figure 6A, the magnitudes of the acupressure force are approximately 3.5 N, 6 N, and 8 N for the light mode, middle mode, and high mode, respectively. Clearly, when the massage force is gradually increased, the compressed displacement on the human tissue also gradually increases. The experimental results, with the record of the representative data shown in Figure 6B, indicated that time for the temperature to reach 41 °C was approximately 100–140 s. The temperature remained at approximately 40–42 °C with small fluctuations, matching the prediction pattern. The LLLT experimental results are given in Table 3. When the minimum power density was 6.5 mW/cm^2^ and the minimum target dose was 4.5 J/cm^2^, the treatment time was around 700 s (11–12 min), which was the designed duration of the treatment.

### 3.2. The Pilot Test Indicated the Treatment Efficacy in Terms of Symptom Relief for Subjects with Knee OA

As shown in Figure 7, the data from eligible 43 volunteers were collected and statistically analyzed at each time point. A lower WOMAC score indicated symptom relief. The total WOMAC score significantly dropped after the first treatment period over 30 days. After the second treatment period over 60 days, the score was significantly lower than at the first assessment point, and it remained stable after the first post-treatment period over 90 days. After two post-treatment periods over 120 days, the total WOMAC score was back to the level of the first assessment point, yet it was still significantly lower than that of the baseline. The scores for pain, stiffness, and functioning presented similar trends, respectively.

## 4. Discussion

Accordingly, the whole hybrid system was established, and the entire structure of the hybrid therapeutic device was assembled. The pilot test results from elderly community-dwelling subjects with KOA demonstrated that the multi-functional hybrid therapeutic device consisting of LLLT, heat therapy, and popliteal fossa massage therapy significantly relieved the subjects’ symptoms. LLLT includes two different types of lasers: a wavelength of 850 nm and a wavelength of 660 nm that can penetrate the skin and work on the inner tissue. Both lasers are recognized as medical-use lasers with suitable wavelengths for KOA treatment, i.e., near-infrared light and infrared light, which provide biochemical changes for cells and tissue [35,36]; thus, LLLT can help tissue regeneration and pain relief [37]. Moreover, the heat therapy in our device based on the control model led to a mild and stable temperature increase to around 41 °C, localized around the popliteal fossa and kneecap, which maintained the best treatment environment and relieved nerve pain [38,39,40,41]. Furthermore, massage therapy focuses on the popliteal fossa and inside the circulatory system, and stimulation could promote conditions for subjective feeling in the popliteal fossa [42].

The WOMAC total score, pain score, stiffness score, and function score were simultaneously improved by the intervention of the CUHK-OA-M2 device over time. It is worth mentioning that, as compared with joint stiffness and function, the pain score was improved and maintained for an extended time, which is important as pain is the most common symptom at the early stage of KOA that severely reduces patients’ life quality [4,5]. Generally, as a subjective outcome, patients tended to be more aware of pain relief than joint stiffness or function. Physiologically, reflected in one’s nerves, pain is also more sensitive than tissue stiffness and joint function [43].

In addition to our findings on LLLT, heat therapy, and massage therapy, the benefits of KOA are presented from existing studies. A consensus from the Society for Translational Medicine only recommended massage therapy sessions and LLLT, but moderately recommended heat therapy for KOA [7]. Compared with massage therapy alone, it is reported that when supplemented with exercise, massage therapy can alleviate pain and improve knee function [22,23,44]. Heat therapy could promote local circulation of the knee joint, where the collagen synthesis induced by the heating signal could further improve functioning, pain tolerance, and muscle matrix metabolism [20,45]. LLLT could activate enzymes in oxidative metabolism [46], suppress inflammation levels [47], and inhibit Aδ/C neuro fibers [16] to relieve joint pain and improve knee function. In this study, the hybrid therapeutic device could perform LLLT, heat therapy, and local massage therapy simultaneously. The combination treatment, by improving local circulation and metabolism, might reduce inflammation symptoms. In addition, for our design and test, the LLLT [48] and massage therapy (refer to Table 2) acted on the popliteal fossa with a moderate stimulation depth, so the therapeutic signals hardly penetrated through the patellar bone and focused on deep tissues to stimulate local circulation and healing. The CUHK-OA-M2 device was specifically designed for treating the popliteal fossa posteriorly (Figure 2A). The novel design structure firmly secured the core’s healing component to keep the laser beam, heating power, and mechanical massage output stable and effective.

There are some limitations in this study. Firstly, we did not include a control group in our pilot test; secondly, we did not include a group with unitary treatment to demonstrate the advantage of the synergistic effects of this multifunctional hybrid therapeutic device; and thirdly, we did not use clinical imaging modalities for classifying KOA stages which are generally required for the clinical definition of KOA for perspective clinical studies.

Based on the therapeutic advantages of the CUHK-OA M2 device, we hypothesized that the integrated treatment could improve local circulation and regulate metabolism. In future studies, we will continue to investigate the clinical outcomes of our devices based on current finding. Moreover, to study the mechanism in the conditions that are promoted by our device, we will carry out more biochemical tests to evaluate the changes in local and systemic inflammation factors in synovial fluid and blood.

## 5. Conclusions

The developed hybrid therapeutic device (CUHK-OA-M2) with intervention effects from the combination of local massage therapy, heat therapy, and low-level laser stimulation effectively relieved the symptoms of community-dwelling elderly subjects with KOA, as confirmed in our pilot human test. Large-scale prospective and randomized clinical trials are recommended for future research to provide evidence for future clinical applications.

## Figures and Tables

**Figure 1 bioengineering-10-00095-f001:**
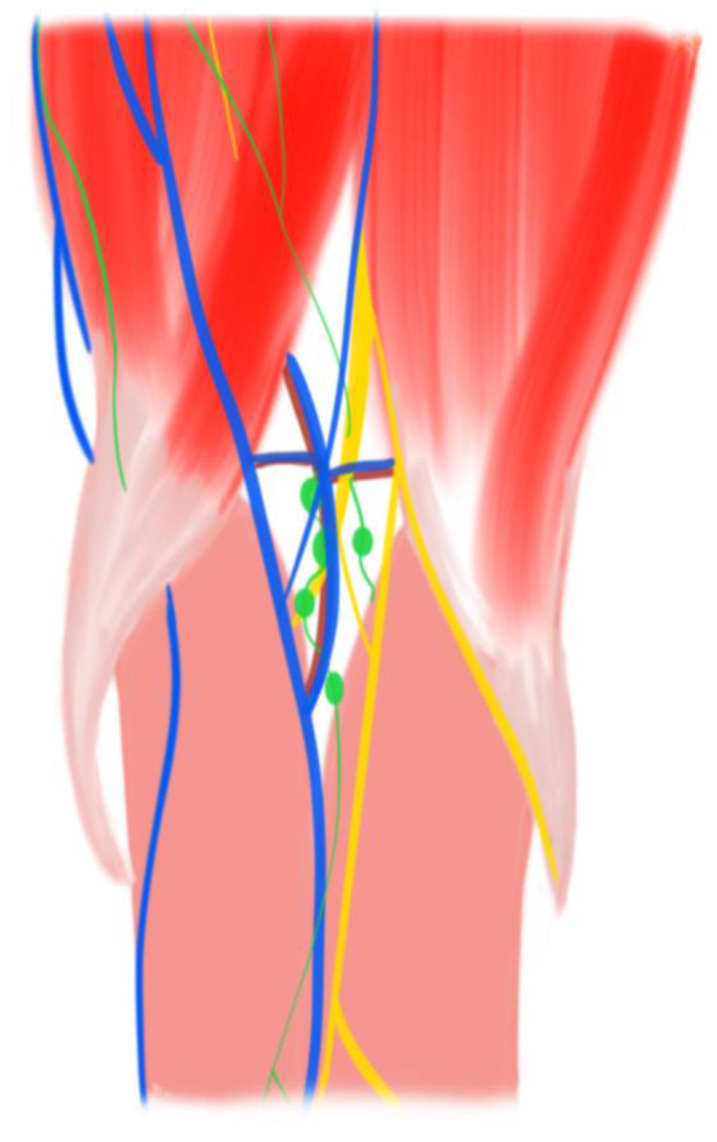
Illustration of popliteal fossa anatomy. Yellow: nerve; blue: vein; red: artery; green: lymph system.

**Figure 2 bioengineering-10-00095-f002:**
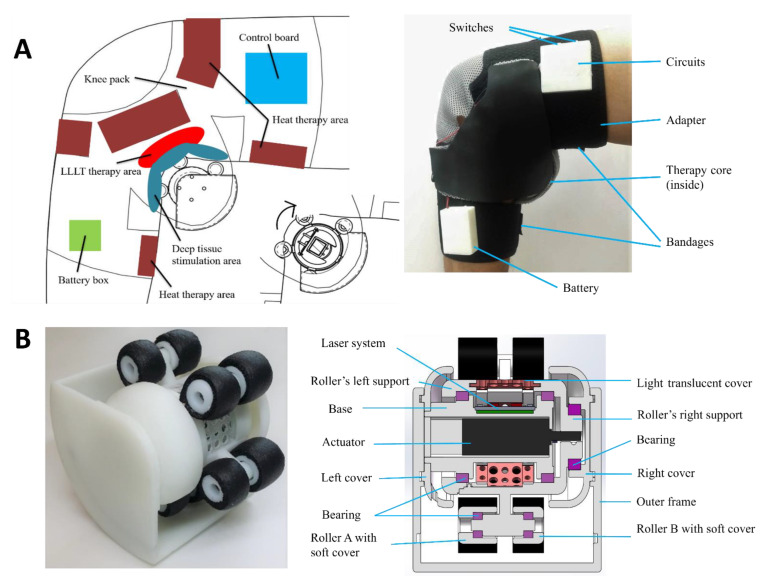
Device design and development: (**A**) Overview of treatment locations and a novel combination therapeutic device CUHK-OA-M2; (**B**) Core treatment components of the device: massage roller, laser emitter, and heat producer.

**Figure 3 bioengineering-10-00095-f003:**
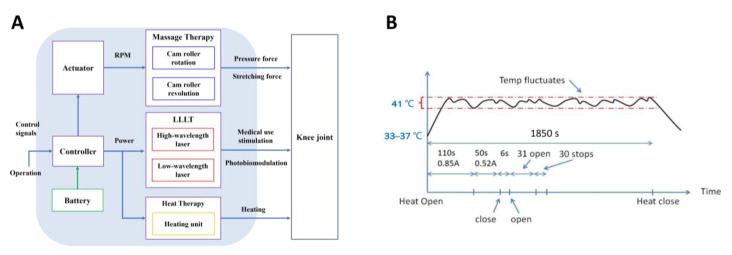
Control system for hybrid therapeutic device. (**A**) Schematic diagram of the therapeutic device system; (**B**) Methodology of the heat therapy system control used to keep the skin surface temperature at 41 degrees centigrade.

**Figure 4 bioengineering-10-00095-f004:**
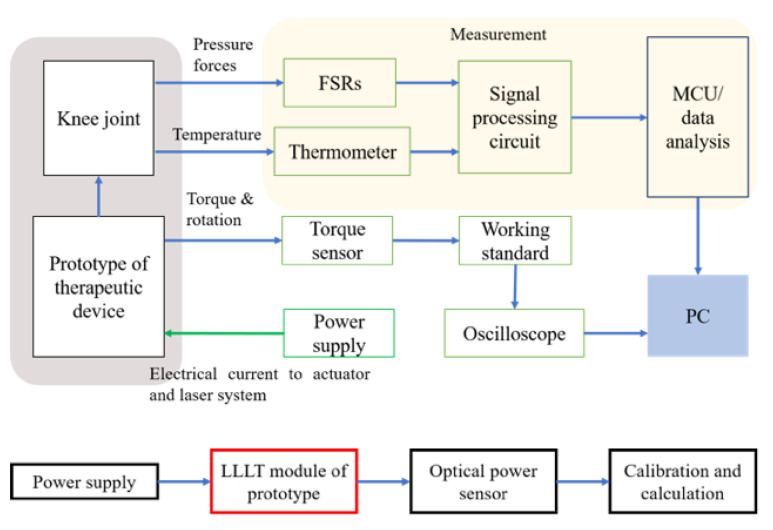
Treatment stimulator test protocols.

**Figure 5 bioengineering-10-00095-f005:**
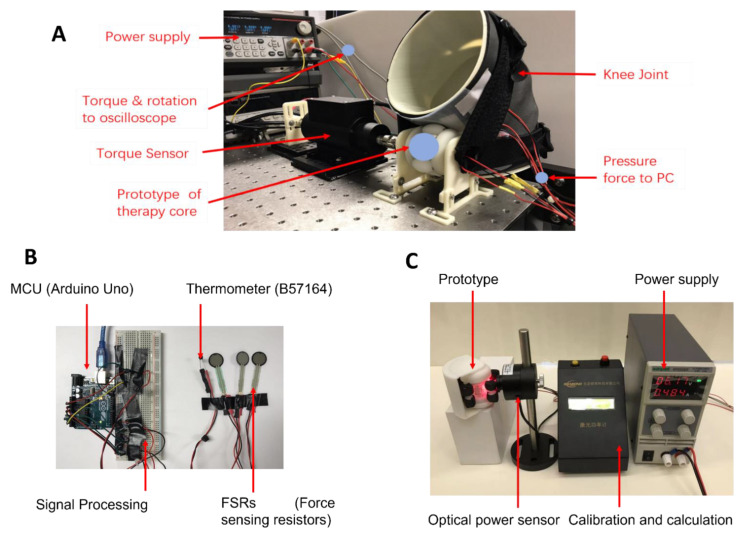
Experimental setup and the measurement system in the research laboratory. (**A**) Experimental bench setup of the hybrid therapeutic device used to test the massage therapy; (**B**) Prototype of the measurement system used to test the massage force and temperature; (**C**) Experimental bench setup of LLLT in the research laboratory.

**Figure 6 bioengineering-10-00095-f006:**
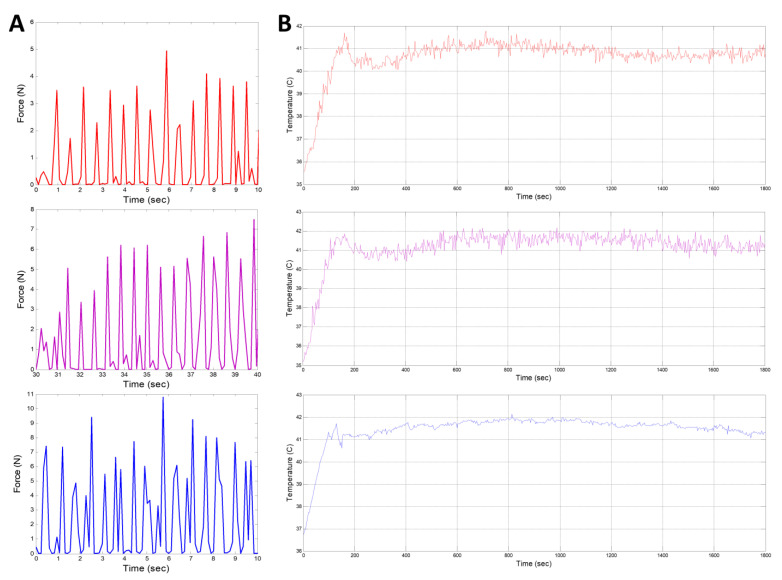
(**A**) Representative curve of massage therapy testing subject; (**B**) Representative curve of thermal testing subject. For (**A**,**B**), top: Light mode; middle: Middle mode; bottom: High mode.

**Figure 7 bioengineering-10-00095-f007:**
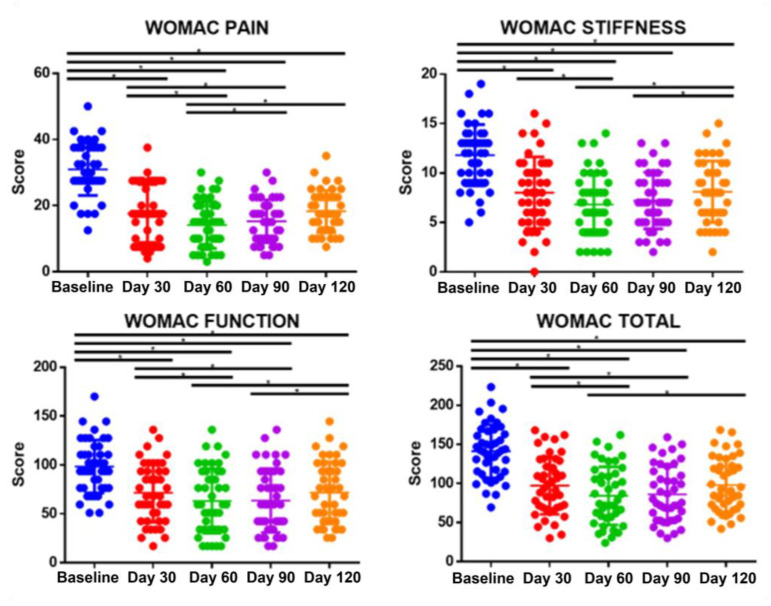
The WOMAC score of each observation point, treatment period: day 30 and day 60; post treatment period: day 90 and day 120; One-way ANOVA, Tukey’s test, *n* = 43, * *p* < 0.05.

**Table 1 bioengineering-10-00095-t001:** Module and physiotherapy system (CUHK-OA-M2).

Module	Components	No.	Stimulation Therapy *
Module 1	Actuator (DC motor, GA6-N20)	1	Massage therapy
	Roller A	5	
	Roller B	5	
Module 2	660 nm laser generator	2	LLLT
	850 nm laser generator	6	
	Heat sink	1	
Module 3	Heating wire (nickel-chromium alloy)	295 cm	HT

* LLLT: low-level laser therapy, HT: heat therapy.

**Table 2 bioengineering-10-00095-t002:** Parameters of deep tissue stimulation CUHK-OA-M2 device.

**Massage Therapy**	
**Parameter**	**Magnitude**
No. of rollers Nr	5
Rotation direction	From posterior to anterior
Rotation speed Sr	20 rpm
Stimulation pulse Mp	100 times/min
Massage pressure	9–30 kPa
Stimulation depth Dp	3–15 mm
**Low-level laser therapy**	
**Parameter**	**Magnitude**
Power density (Peak)	20 mW/cm^2^
Power density (Effective)	5 mW/cm^2^
Dose	3–20 J/cm^2^
Irradiation area	5–10 cm^2^
Laser wavelength	660 nm and 850 nm
**Heat therapy**	
**Parameter**	**Magnitude**
Temperature	40–42 degrees centigrade

**Table 3 bioengineering-10-00095-t003:** LLLT experimental results.

	Irradiance mW/cm^2^		
Measurement Point	Left Side	Middle	Right Side
Experiment	6.6	19.1	6.5

## Data Availability

Not applicable.

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
