# Peer review of "Hybrid Therapeutic Device (CUHK-OA-M2) for Relieving Symptoms Induced by Knee Osteoarthritis"

_bioengineering, 2023, doi:10.3390/bioengineering10010095_

Round 1

Reviewer 1 Report

Dear Authors,

The paper presents a hybrid therapeutic device for releasing the symptom induced by knee osteoarthritis. The research looks interesting and the findings presented in the paper could be useful in medical applications. However, the quality of the paper should be improved before publication. Please see below my comments and suggestions.

1. I feel the title of the paper is long and confusing. It's better to update the title of the paper.

I suggest a title like this: "Hybrid Therapeutic Device (CUHK-OA-M2) for Releasing the Symptom Induced by Knee Osteoarthritis". As you are proposing a hybrid system by combining different assistive techniques, the suggested title seems suitable for the paper.

2- Abstract: Please revise the abstract and remove the following terms: "Objective:", "Methods and Materials:", "Results:", and "Conclusions:".

3-Keywords: I suggest adding the following keyword to the paper: "Physical Stimulation".

4- Subsection 2.1 heading: "2.1. The design of novel combination medical device" seems not suitable. You can use a better term. For example, I suggest using "2.1. Design of the hybrid therapeutic device".

5. Please improve the quality of Figure 2. The text is not readable.

6. Line 96: the provided information on the laser modules used in the device is not completed. For example, what is the output power of the laser modules used? what is the manufacturer part number of the modules? The method section needs to be elaborated and the details of the components used in the device should be included.

7. The paper claims that the proposed system is a novel device and the whole system is designed and developed. However, the circuit design of the control system is not included in the paper. Please revise the method section and include the above-mentioned information.

8- Regarding thermal therapy and heat system: How the temperature is controlled? Mechanical design is illustrated in figure 2, but the electrical design (circuit) is not presented.

9- Subsection "2.2. Lab experiment setup for massage therapy, HT and LLLT". Please include the experimental stand and the device developed.

10- Language edit and moderate English changes required. Different issues are existing in the text.

Author Response

Dear Editors and Reviewers,

Thank you for inviting us to revise our manuscript (bioengineering-2063862). Enclosed please find our point-by-point reply based on the comments made by your review team with recommendation for minor revision. We really appreciate your recommendation and constructive comments. Changes or modifications made in the attached version are highlighted using the “Tracking changes” function for easy reference for both response letter and our revised manuscript.
Hope that our replies and revised manuscript would be satisfactory for publication in Bioengineering.

Best regards,

Professor Ling Qin (on behalf of all co-authors)

Response to Comments of Reviewer 1

The paper presents a hybrid therapeutic device for releasing the symptom induced by knee osteoarthritis. The research looks interesting and the findings presented in the paper could be useful in medical applications. However, the quality of the paper should be improved before publication. Please see below my comments and suggestions.

Response to general comments: The authors appreciate the encouragement to and confirmation for our original contribution to developing our hybrid therapeutic device for knee OA. The authors have made point-by-point replies below to address all constructive comments and suggestions as well as critics to our work together with improvement of paper quality.

Reviewer #1: Point 1: I feel the title of the paper is long and confusing. It's better to update the title of the paper.

I suggest a title like this: "Hybrid Therapeutic Device (CUHK-OA-M2) for Releasing the Symptom Induced by Knee Osteoarthritis". As you are proposing a hybrid system by combining different assistive techniques, the suggested title seems suitable for the paper.

Response 1:

Thanks for the constructive suggestion on the modification of paper title and we have revised the title as suggested in Page 1 Lines 1-4, initially as “A Wearable Biomechanically-Designed Medical Device (CUHK-OA-M2) Integrated Phototherapy, Thermal Therapy and Mechanical Stimulation to Release the Symptom Induced by Knee Osteoarthritis” and now in Page 1 Lines 1-2 of the revised version as “Hybrid Therapeutic Device (CUHK-OA-M2) for Releasing the Symptom Induced by Knee Osteoarthritis”

Reviewer #1: Point 2: Abstract: Please revise the abstract and remove the following terms: "Objective:", "Methods and Materials:", "Results:", and "Conclusions:".

Response 2:

Thanks for the advice and we have revised the abstract by removing "Objective:", "Methods and Materials:", "Results:", and "Conclusions:" in Page 1 Line 13, Page 1 Line 17, Page 1 Line 23, and Page 1 Line 27.

Reviewer #1: Point 3: Keywords: I suggest adding the following keyword to the paper: "Physical Stimulation".

Response 3:

Thanks for the suggestion and we have added the keyword “physical stimulation” to the revised version on Page 1 Lines 30-31.

Reviewer #1: Point 4: Subsection 2.1 heading: "2.1. The design of novel combination medical device" seems not suitable. You can use a better term. For example, I suggest using "2.1. Design of the hybrid therapeutic device".

Response 4:

Thanks for the suggestion and we now have revised it accordingly as “2.1. Design of the hybrid therapeutic device” in Page 3 Line 80 of the revised manuscript.

Reviewer #1: Point 5: Please improve the quality of Figure 2. The text is not readable.

Response 5:

Many thanks for the suggestion and point out that figure 2 was not readable in our initial submission. Now we have updated Figure 2 with high resolution in Page 3 Line 89 of our resubmitted version.

Reviewer #1: Point 6: Line 96: the provided information on the laser modules used in the device is not completed. For example, what is the output power of the laser modules used? what is the manufacturer part number of the modules? The method section needs to be elaborated and the details of the components used in the device should be included.

Response 6:

Thanks for the comment and suggestion. Yes, the types of LLLT components for laser modules are important to present in method section, so the reader can clearly know the relationship between the targeted LLLT doses and the laser generators. Now we have supplemented relevant information of the types of laser generator in Page 3 Lines 96-97 as “two types of laser models with distinct wavelengths: 660 nm (T2835, output power: 0.4W) and 850 nm (T2835, output power: 1.2W);”. Since the targeted LLLT requirements (power density and dose) are listed in Table 2, so we could calculate the required components accordingly.

Reviewer #1: Point 7: The paper claims that the proposed system is a novel device and the whole system is designed and developed. However, the circuit design of the control system is not included in the paper. Please revise the method section and include the above-mentioned information.

Response 7:

Many thanks for the important comment and suggestion. Since the treatment for this paper is our key component, some contents were not discussed in our initial version as pointed out. Now, we have supplemented relevant information as “2.2. Control system of the hybrid therapeutic device”, and the control system for the hybrid therapeutic device has been introduced in Page 4 Lines 103-112 of the revised version: “The principle of the whole therapeutic device system and the application procedure are summarized in Figure 3A. Firstly, the user received our combination treatment by wearing the proposed therapeutic device where the user could adjust the adapter of the therapeutic device to reach a satisfactory degree of tightness and start the treatment of the device. Then, based on the targeted speed, torque and pressure force of the massage therapy, the controller sent the appropriate voltage signal to the actuator. The temperature was maintained within 40 to 42 degrees centigrade by controlling the output current magnitude. Finally, the system provided sufficient optical power to the popliteal fossa to meet the therapeutic power density and dose value.”

(Please find the figure in revised manuscript)

Figure 3. Control system for hybrid therapeutic device. A. Schematic diagram of the therapeutic device system; B. Methodology of the HT system control used to keep the skin surface temperature at 41 degrees centigrade.

Reviewer #1: Point 8: Regarding thermal therapy and heat system: How the temperature is controlled? Mechanical design is illustrated in figure 2, but the electrical design (circuit) is not presented.

Response 8:

Thanks for the comment. Heat therapy control is to enable the temperature to reach human skin suitably, so the control method for heat therapy was discussed in our initial version. We used an on-off control method to control the temperature on human skin to keep the temperature around 41 degrees centigrade. Now, this part has been added as suggested in Page 4 Lines 113-124 of our resubmitted version: “As shown in Figure 3B, in order to consume less electrical power in the warming system, we used a high current value at the beginning until the temperature of the skin surface reached  and then used the second gear of the current level for the rest of the treatment. To avoid the temperature from exceeding , which was out of the range of combination treatment, the off-control operations are required during the treatment of the warming system as shown in Figure 3B. With this on-off control methodology, the temperature of the skin surface could be maintained around  with suitable fluctuation.”

(Please find the figure in revised manuscript)

Reviewer #1: Point 9: Subsection "2.2. Lab experiment setup for massage therapy, HT and LLLT". Please include the experimental stand and the device developed.

Response 9:

Thanks a lot for the suggestion. The lab experimental setup and its development have been now supplemented in Page 6 Lines 136-152 of the revised version as follows:

“2.4. Experimental stand and development

The experimental setup of the hybrid therapeutic device was established in the research laboratory as shown in Figure 5. The acupressure force could be changed by adjusting the bandage around the knee joint and the force sensor and torque sensor determined the relationship between the preload and massage force. As shown in Figure 5B, the prototype of the measurement system was developed. The thermometer (B57164) is a kind of thermistor that responds to the temperature in real time. The force-sensitive resistor (FSR 402) is a unit to measure the pressure force and translate it into an electrical signal, and the MCU (Arduino Uno) is used to analyze the signals from the sensors. LLLT test stand as shown in Figure 5C, the sensor (VLP-2W, Beijing Ranbond Technology Co., Ltd.) was used for laser power experiment.

(Please find the figure in revised manuscript)

Figure 5. Experimental setup and the measurement system in the research laboratory. A. Experimental bench setup of the therapeutic device used to test the massage therapy; B. Prototype of the measurement system used to test the massage force and temperature; C. Experimental bench setup of LLLT in the research laboratory.”

Reviewer #1: Point 10: Language edit and moderate English changes required. Different issues are existing in the text.

Response 10:

Thanks for the comment and advice. The authors sought language assistance from an editor of an SCI journal for our resubmitted version.

Reviewer 2 Report

The research work proposed by Li ZOU and entitled "A Wearable Biomechanically-Designed Medical Device (CUHK-OA-M2) Integrated Phototherapy, Thermal Therapy and Mechanical Stimulation to Release the Symptom Induced by Knee Osteoarthritis" consists of proposing and developing a new combined therapy tool integrating the mechanical stimulus for the treatment of the symptoms of knee osteoarthritis. this method has been tested, optimized and validated on 50 patients. The results of this method of treatment are very interesting, they significantly reduce knee pain, according to the authors.

The study is very interesting, the research theme too and the results are relevant. this paper goes very well with the newspaper scoup.

The article in general is very well structured and written.

1. the abstract can be improved and in particular the first objective part.

2. In the method section, I ask the authors to specify when this method is the most effective in relation to the state of osteoarthritis. can it be recommended for advanced stages of osteoarthritis or it must be at the beginning.

3. Can this mothodoligy be the subject of numerical modeling to test and optimize other parameters and simulate other configurations easily.

4. The work presented, although very important, is more engineering and innovation than research. I ask the authors to enrich the discussion and the conclusion by explaining more scientifically the reason for these results.

Author Response

Dear Reviewer,

Thank you for inviting us to revise our manuscript (bioengineering-2063862). Enclosed please find our point-by-point reply based on the comments made by your review team with recommendation for minor revision. We really appreciate your recommendation and constructive comments. Changes or modifications made in the attached version are highlighted using the “Tracking changes” function for easy reference for both response letter and our revised manuscript.
Hope that our replies and revised manuscript would be satisfactory for publication in Bioengineering.

Best regards,

Professor Ling Qin (on behalf of all co-authors)

Response to Reviewer 2 Comments

The research work proposed by Li ZOU and entitled "A Wearable Biomechanically-Designed Medical Device (CUHK-OA-M2) Integrated Phototherapy, Thermal Therapy and Mechanical Stimulation to Release the Symptom Induced by Knee Osteoarthritis" consists of proposing and developing a new combined therapy tool integrating the mechanical stimulus for the treatment of the symptoms of knee osteoarthritis. this method has been tested, optimized and validated on 50 patients. The results of this method of treatment are very interesting, they significantly reduce knee pain, according to the authors.

The study is very interesting, the research theme too and the results are relevant. this paper goes very well with the newspaper scoup.

The article in general is very well structured and written.

Response to general comments: The authors appreciate the encouragement to and confirmation for our original contribution from innovation to its clinical validation.

We have made point-by-point replies below to address all constructive comments and suggestions as well as critics to our work.

Reviewer #2: Point 1: The abstract can be improved and in particular the first objective part.

Response 1:

Thanks a lot for the comment and suggestions. As advised, we modified our study objective as “This study aims to develop a novel hybrid therapeutic device that can meet various requirements for knee therapy. Our hybrid therapeutic device (CUHK-OA-M2) integrated with low-level laser therapy, heat therapy, and local massage therapy can help patients with KOA to release their clinical symptoms effectively.” in Page 1 Lines 13-29 of the revised version.

Reviewer #2: Point 2: In the method section, I ask the authors to specify when this method is the most effective in relation to the state of osteoarthritis. can it be recommended for advanced stages of osteoarthritis or it must be at the beginning.

Response 2:

Many thanks for the comment and question. Since the physical stimulations are very suitable for early-stage KOA, so this therapeutic device is recommended for early-stage KOA (Stages 1-3) as this has also been validated in the recruited patients. We are also interested to know if our device would also be helpful for patients with late stage of KOA but subject to future clinical study with or without adjustment of the physical parameters used in our device. We revised the method section in Page 3 Lines 81-82 “we designed an entire biophysical system (CUHK-OA-M2) for KOA in the popliteal fossa in patients with early-stage KOA [4,6])”, and discussed in Page 9-10 Lines 239-246. For advanced stages of OA, surgery might be needed and physical stimulations could be used as the supplementary treatment.

Reviewer #2: Point 3: Can this methodology be the subject of numerical modeling to test and optimize other parameters and simulate other configurations easily.

Response 3:

Thanks a lot for the comment and question. Indeed, this is valuable for exploration. We will analyze the results by developing a system to evaluate the conditions of the patient and to optimize the parameters in the future. What we can think of at this stage: depending on the test results to be obtained, we will try to use a polynomial equation to fit the results to predict the changing trend of each parameter. A subject could estimate the progression of conditions after following the suggested course of treatment by using our model, and from the prediction result we could optimize the parameters for the subject. And this stage, we still need to develop a smart system to evaluate the symptoms and estimate the rehabilitation progress. As shown in the following figure, the sensing system and user parameters are added; H is the hydrops of the knee joint, S is the tissue stiffness, O is the other parameters, and the user parameters include gender. These parameters are personalized settings to make this system more accurate. First, from the questionnaire survey, the user can input his or her personal information and the 1st WOMAC result; meanwhile, the H, S, and O parameters are detected from the sensing system. Then, the smart system determines the stage of the user: G0 is healthy, G1 to G3 are our target subjects, and G4 is the end stage of a patient, who will be asked to go to a hospital. Second, the stimulation treatment suggestion (DTS: massage, LLLT: low-level laser therapy, HT: heat therapy) and the predicted WOMAC results are provided after evaluating the symptoms of the user. After following the suggestion, the user is treated by the therapeutic device, and the system can estimate the WOMAC result depending on the actual usage; then, from a comparison with the predicted WOMAC result, the evaluation model can be modified. The users may also provide feedback for the WOMAC result to be tested to help the system modify the parameters of the assessment algorithm.

(Please find the figure in the attached file)

The workflow of the smart system we are now proposing to analyze and control the rehabilitation progress of the therapeutic device.

Reviewer #2: Point 4: The work presented, although very important, is more engineering and innovation than research. I ask the authors to enrich the discussion and the conclusion by explaining more scientifically the reason for these results.

Response 4:

Thanks a lot for the constructive suggestion and comment. As commented, engineering and innovation are very important for this study and future research. So, we enhanced our discussions on the engineering design and physical stimulations design in Discussion Section to explain more about the lab experimental results and pilot human results with potential underlying mechanism of the improvement of symptom release in Page 9 Lines 224-238 of the resubmitted version “Accordingly, the whole hybrid system was established, and the entire structure of the hybrid therapeutic device was assembled. The pilot test results performed in elderly dwelling subjects with KOA demonstrated that the multi-functional hybrid therapeutic device consisting of LLLT, heat therapy, and popliteal fossa massage therapy could significantly release the subjects’ symptoms. LLLT includes two different types of lasers: a wavelength of 850 nm and a wavelength of 660 nm that can penetrate the skin and work on the inner tissue. Both lasers are recognized as medical-use lasers with the suitable wavelengths for KOA treatment, i.e., near-infrared light and infrared light, which provides biochemical changes for cells and tissue [36-37], so the LLLT can help tissue regeneration and pain relief [38]. Besides, the heat therapy in our device based on the control model leads to a mild and stable temperature increase to around  localized around popliteal fossa and kneecap, which maintains the best treatment environment and relieves nerve pain. [39-42] Furthermore, massage therapy focuses on the popliteal fossa and inside the circulatory system, and the stimulation could promote the conditions of subjective feeling in the popliteal fossa [43].”

And the details of the discussion are revised in the Section “Discussion” (Pages 9-10, Lines 223-277).

And in this study, the therapeutic area of the knee joint is determined, and the mechanical structures, considering ergonomics for the knee joint and popliteal fossa, are designed. A combination treatment involving massage, LLLT and heat therapy is designed, and the doses and parameters of each stimulation are calculated. A prototype of the therapeutic device is fabricated and tested. The massage result is quantized, and the LLLT system is designed and tested. Finally, a course of treatment and measurement (WOMAC OA index) for patients with KOA is designed for the pilot test.

[36] Chung H., Dai T., Sharma S. K., Huang Y. Y., Carroll J. D., & Hamblin M. R. The nuts and bolts of low-level laser (light) therapy. Annals of Biomedical Engineering 2012; 40: 516–533.

[37] Cotler H. B., Chow R. T., Hamblin M. R., & Carroll J. The use of low level laser therapy (lllt) for musculoskeletal pain. MOJ Orthop Rheumatol 2015; 2(5): 00068.

[38] Farivar S., Malekshahabi T., & Shiari R. Biological effects of low level laser therapy. J Lasers Med Sci. 2014; 5(2): 58-62.

[39] Gao Z., Guo X., Chen J., and Duan C. Hyaluronic acid inhibited the upregulation of heat shock protein 70 in human chondrocytes from osteoarthritis and Kashin-Beck disease. Biocell 2019; 43(2): 99-102.

[40] Hojo T., Fujioka M., Otsuka G., Inoue S., Kim U., Kubo T. et al.. Effect of heat stimulation on viability and proteoglycan metabolism of cultured chondrocytes: preliminary report. Journal of Orthopaedic Science 2003;8(3): 396-399.

[41] Ito A., Aoyama T., Tajino J., Nagai M., Yamaguchi S., Iijima H., Kuroki H., et al.. Effects of the thermal environment on articular chondrocyte metabolism: a fundamental study to facilitate establishment of an effective thermotherapy for osteoarthritis. J Jpn. Phys. Ther. Assoc. 2014;17: 14-21.

[42] Son Y., Kim H., Choi W., Chun C., and Chun J. RNA-binding protein ZFP36L1 regulates osteoarthritis by modulating members of the heat shock protein 70 family. Nature Communications 2019;10:77.

[43] Ko T., Lee S., & Lee D. Manual therapy and exercise for oa knee: effects on muscle strength, proprioception, and functional performance. Journal of Physical Therapy Science 2009; 21: 293-9.

Round 2

Reviewer 1 Report

Dear Authors,

Thank you for revising the paper and addressing the comments.

The paper could be suggested for acceptance.